# Awareness, attitudes, and beliefs about palliative care: Results from a representative survey of the Italian-speaking Swiss population

**Nicola Diviani**[1,2]*, **Marco Bennardi**[1,2], **Claudia Gamondi**[3¤], **Piercarlo Saletti**[4], **Georg Stüssi**[5,6], **Michel Delbue-Luisoni**[7], **Sara Rubinelli**[1,2]

1 Swiss Paraplegic Research, Nottwil, Switzerland, 2 Faculty of Health Sciences and Medicine, University of Lucerne, Lucerne, Switzerland, 3 Palliative and Supportive Care Clinic, Oncology Institute of Southern Switzerland, Ente Ospedaliero Cantonale, Bellinzona, Switzerland, 4 Gruppo Ospedaliero Moncucco, Lugano, Switzerland, 5 Clinic of Hematology, Oncology Institute of Southern Switzerland, Ente Ospedaliero Cantonale, Bellinzona, Switzerland, 6 Faculty of Biomedical Sciences, Università della Svizzera Italiana, Lugano, Switzerland, 7 Palliative ti, Camorino, Switzerland

¤ Current address: Palliative and Supportive Care Service, Lausanne University Hospital and University of Lausanne, Lausanne, Switzerland
* Nicola.Diviani@paraplegie.ch

**Data Availability Statement:** All data files are available on the figshare data repository: Diviani, Nicola; Bernardi, Marco; Gamondi, Claudia; Saletti,

## Abstract

### Objective

To understand the knowledge and awareness of palliative care in the Italian-speaking Swiss general population, describing main misconceptions or false beliefs and their relationship with attitudes towards palliative care.

### Methods

Cross-sectional representative population survey (N = 313).

### Results

We observed a high awareness of «palliative care,» although it is mainly associated with pain management and the very last days of life. While false beliefs are relatively rare, there is low awareness of goals, targets, and services offered by palliative care. Overall the Italian-speaking Swiss population has a good predisposition towards palliative care, but negative attitudes are more common among those who lack knowledge. More than one-third of respondents are interested in receiving more information about palliative care, especially from their healthcare providers or through dedicated information points.

### Conclusion and practice implications

Health communication interventions to promote palliative care are needed because there is still significant unclarity about the goals of palliative care, which negatively affects its acceptance. This study instructs on how to intervene specifically in the Italian-speaking part of

Piercarlo; Stüssi, Georg; Delbue - Luisoni, Michel; et al. (2022): 210109_Palliativ TI.sav. figshare. Dataset. https://doi.org/10.6084/m9.figshare.21762824.v1.

**Funding:** Funding for this research was awarded to SR, ND, CG, GS, and PS under a grant, which was financed by Swiss Cancer Research (www.krebsforschung.ch; Grant number KFS-4163-02-2017). In addition, SR, ND, CG, GS, and PS received financial support for the conduction of the survey by Palliative-ti (www.palliative-ti.ch; Grant number N/A). The funders had no role in study design, data collection and analysis, decision to publish, or preparation of the manuscript.

**Competing interests:** The authors have declared that no competing interests exist.

Switzerland, including what to communicate and how. Further, our findings can inspire similar studies in other Swiss regions or countries that can optimize recognition, knowledge, and understanding and contribute to filling gaps in populations' health service demand and utilization.

## Introduction

Palliative care is defined as "an approach that improves the quality of life of patients and their families who are facing problems associated with a life-threatening illness by preventing and relieving suffering through the early identification, correct assessment, and treatment of pain and other problems, whether physical, psychosocial or spiritual" [1]. In Switzerland, like many other developed countries, most deaths nowadays are caused by chronic, progressing diseases such as cancer, cardiovascular diseases, or respiratory diseases that often demand a high need of care for a prolonged time before death [2]. Also, in the following decades, it is projected that the portion of older adults worldwide will continue to grow [3]. The number of people in need of palliative care will, therefore, significantly increase, making it crucial to foster its integration into health care systems, ensuring that a growing number of people can profit from it in the future [4].

Despite their potential, the uptake of palliative care services is still suboptimal. Among the several obstacles to palliative care utilization described in the literature, it is now established that, especially in high-resource healthcare systems, a lack of awareness about its characteristics among the public plays a central role [5]. Studies conducted in different countries have shown insufficient knowledge and understanding of palliative care in the general public. Estimates of the share of people without knowledge of palliative care range from up to 40% in Europe to up to 60–70% in the United States, Korea, and India [6–13].

In addition, studies have shown that people who have heard of palliative care often only have a vague idea about it. Specifically, many do not precisely know what the objectives of palliative care are, and misconceptions are widespread [14]. For instance, many reported automatically thinking of death when hearing the term. They are convinced that accepting palliative care requires stopping other treatments and associating palliative care with giving up. They believe it is strictly related to end-of-life care or only for older adults. palliative care is also often equated with symptom management, with many unaware that it includes psychological and spiritual support, bereavement services, and support for family and caregivers [15–17]. Other common misconceptions are that palliative care is some "alternative care therapy" or "placebo" or that it is confounded with hospice care [12, 18–20].

Evidence suggests that this lack of awareness and knowledge about palliative care cannot be attributed to specific population groups, but it is relatively widespread at all levels of society. However, some factors are associated with better knowledge of palliative care, such as being of the female sex, being older (40+), having higher education, having a friend or a relative needing palliative care, or working in a health care setting [14].

Lack of awareness and misconceptions about palliative care are not harmless. They can hinder patients from being referred to palliative care at the right time and lead to underusing services. Lack of knowledge of palliative care slows timely referrals and restricts informed decision-making. It also disrupts the goals of care conversation and makes patient-centered care less effective. The reasons for this are manifold. Providers might fear that referring patients for palliative care may result in a loss of hope or may lead the patient to believe their

provider is giving up on them. Providers might reserve palliative care referrals for care related to a terminal stage of the illness or symptom management [15]. Also, misperceptions about palliative care could hinder people from having these services. For example, misperceptions stating that palliative has to do only with end-of-life care or hospice or that it has to do with elder care could make people believe that these services are not appropriate for friends, loved ones, or even themselves [16].

The main objective of this paper is to find evidence to inform tailored communication interventions to foster the use of palliative care services in the Italian-speaking part of Switzerland. The present study is part of a larger project aimed at exploring the discourse around palliative care. Specifically, we aimed at i) understanding the level of knowledge and awareness of palliative care, ii) describing main misconceptions or false beliefs and their relationship with attitudes towards palliative care, and iii) identifying the segments of the population most in need of dedicated interventions and best ways to reach them.

## Materials and methods

### Sample

Data were collected in June 2021 through a representative online survey. The LINK Internet Panel, a group of pre-recruited individuals who have agreed to participate in online market research surveys and studies, was utilized for conducting our survey. This panel has a base of over 100,000 individuals, sampled according to quotas of gender and age across Switzerland. It facilitates the collection of data that mirrors the diversity of the general population. To be eligible for our study, individuals were required to live in the Italian-speaking part of Switzerland, be fluent in Italian, and be 18+. A total of 313 individuals participated in the survey.

### Instrument and measures

We developed a questionnaire including a combination of open-ended and closed-ended questions. The questionnaire was divided into six main sections that assessed individuals' (i) knowledge and awareness of palliative care, (ii) beliefs about palliative care, (iii) attitudes towards palliative care, (iv) health-related information-seeking behavior and preferences, and v) socio-demographics.

**Knowledge and awareness of palliative care.** First, respondents were asked whether they had heard of "palliative care." Those who responded affirmatively were subsequently asked in an open-ended question to describe the objectives of palliative care. Additionally, participants were asked to rate their knowledge of the topic on a scale ranging from 1 = 'Not at all knowledgeable' to 7 = 'Very knowledgeable.' Additionally, participants were asked to indicate whether they had direct (personal) or indirect (e.g., through a family member or a friend) experience with palliative care services and whether they had a medical background or education. Respondents who had never heard of the term were redirected to the section of the questionnaire about health information seeking.

**Beliefs.** Participants were presented with a list of 13 statements about palliative care. Seven of these statements were facts (e.g., "Palliative care services include psychological, social, and spiritual support"), and six of them were false beliefs (e.g., "Palliative care shortens life"). The statements were chosen based on established definitions of palliative care, e.g., by the World Health Organization [1], and on common myths surrounding palliative care described in the literature. Participants were asked to evaluate each statement on a 7-point scale ranging from 1 = "Certainly false" to 7 = "Certainly true."

**Attitudes towards palliative care.** Three items measured attitudes toward palliative care. First, participants were asked to rate the utility of palliative care on a 7-point scale ranging

from 1 = "Not at all useful" to 7 = "Very useful." Subsequently, they were asked to rate the like-lihood that they would use palliative care services themselves or recommend them to a friend on a 7-point scale ranging from 1 = "Certainly not" to 7 = "Definitely." The three items were merged in a sum score to be used in multivariate analyses (Cronbach's $\alpha$ = .886, Mean = 17.4, SD = 3.49).

**Health-related information-seeking, ability, and preferences.** Participants were asked to indicate all the sources of health-related information they used. In addition, a single item was used to assess the respondents' level of health literacy. Here, respondents were asked to rate their confidence in finding health-related information on a 5-point scale ranging from 1 = "Very difficult" to 5 = "Very easy." Finally, respondents were asked whether they would be interested in receiving more information about palliative care and through which channels.

**Socio-demographics.** Respondents were asked to indicate their gender, age, highest educational attainment, nationality (Swiss vs. foreign), and religiosity (yes/no). In addition, they were asked to indicate whether they already had a direct or indirect experience with palliative care services and whether they had a medical education or background.

## Data analysis

Data were analyzed both quantitatively and qualitatively. Quantitative variables were analyzed using IBM SPSS Statistics version 21.0. Chi-square tests (for categorical variables) and one-way Analysis of Variance (for continuous variables) with Tukey's post hoc test were conducted to identify group differences among respondents regarding their socio-demographic characteristics. Multivariate regression analyses were used to explore the association of knowledge and beliefs about palliative care with attitudes, controlling for the socio-demographic characteristics of the participants. Responses to open-ended questions were analyzed using qualitative thematic analysis, where we coded the data to identify significant patterns and distilled these into key themes that captured the essence of participants' perspectives. This process was guided by established methods to ensure a rigorous and reliable interpretation of the data [21].

## Ethical considerations

According to the Swiss Federal Act on Research involving Human Beings (Human Research Act, HRA, September 30th, 2011), research not concerning diseases or the structure or function of the human body does not require formal approval from an ethical review board. The Ethical Committee of Canton Ticino (decision number req-2018-00227) confirmed that this applies to the present study. Upon starting the online questionnaire, all participants were informed of the nature and aims of the study and that they could withdraw their consent to participate at any time. Written informed consent was obtained electronically from all participants.

## Results

A total of 313 people participated in the survey. Male and female participants were equally represented in the sample, and the mean age of the sample was slightly below 50 years. Almost two-thirds of the respondents had a lower educational level (high school/professional diploma or less). Most respondents were Swiss, although many also held another citizenship. Less than half of them reported being religious. Most respondents reported no previous experience with palliative care. Among those with experience, the majority had an indirect one, e.g., through a family member or a friend. Around one in ten respondents had an education or a background in a health-related field. More details about the sample characteristics can be found in Table 1.

**Table 1. Sample characteristics.**

| | | All respondents, N = 313 |
|---|---|---|
| | | n (%) |
| **Gender** | Male | 154 (49.1) |
| | Female | 159 (50.9) |
| **Age** | Mean (SD) | 48.9 (15.1) |
| **Education** | Lower | 171 (57.44) |
| | Higher | 127 (42.6) |
| **Nationality** | Swiss | 288 (91.9) |
| | Other | 25 (8.1) |
| **Religiosity** | Yes | 145 (46.3) |
| | No | 117 (37.3) |
| | No response | 51 (16.4) |
| **Experience with palliative care services** | Yes, direct | 31 (10.1) |
| | Yes, indirect | 90 (29.5) |
| | No | 184 (60.4) |
| **Medical background / education** | Yes | 28 (9.0) |
| | No | 285 (91.0) |

## Knowledge and awareness of palliative care

Most respondents reported having heard of the term 'palliative care' (91.9%). Having heard of palliative care was more frequent among older respondents ($p < .01$), while no differences were observed among males and females or people with different educational levels. When asked in an open-ended question to describe the objectives of palliative care, a more complex picture emerged. Although not completely wrong, most answers to this question were overall partial or imprecise. For instance, only a few respondents mentioned the psychological or spiritual component of palliative care or that it is also directed at family members. In many answers, the focus was on pain management and the very last days of the life of a patient with a terminal disease. Only very few respondents, on the other hand, gave utterly wrong answers.

Overall, respondents rated their knowledge of palliative care relatively low, with a mean score of 3.4 (SD = 1.3) on a 7-point scale. Self-rated knowledge was shown not to be associated with any of the socio-demographics.

## Beliefs about palliative care

Participants were presented with a list of common true and false beliefs about palliative care. An overview of answers is presented in Table 2. Among the true beliefs, somehow in contrast with the answers to the open question, participants more often correctly believed that improving quality of life is among the objectives of palliative care (90%), that palliative care offers social, psychological, and spiritual support to the patient (81.5%), that palliative care is also offered at the patient's home (81.1%), and that palliative care is provided by a team of different specialists (80.6%). Around two out of three respondents (67.5%) also correctly believed that palliative care gives a lot of importance to the preferences of the patient. Less than half of respondents correctly believe that the costs of palliative care are covered by basic health insurance (45%) and that palliative care is also for the patient's family members (37.5%).

Among false beliefs, most participants correctly did not believe that palliative care shortens a person's life (77.6%) and that it is only for cancer patients (74.5%). Only around one in two participants correctly did not believe that palliative care is only about pain management

Table 2. Knowledge, beliefs, and attitudes about palliative care.

| | | All respondents, N = 313 | |
|---|---|---|---|
| | | Mean (SD) | n (%) |
| **Ever heard of palliative care** | Yes | | 288 (91.9) |
| | No | | 25 (8.1) |
| **Self-rated knowledge of palliative care** | 1 = No knowledge; 7 = Very much knowledge | 3.4 (1.3) | |
| **Beliefs** | | Mean (SD) | n (%) correct[1] |
| Improving patients' quality of life is one of the objectives of palliative care. | 1 = Certainly false; 5 = Certainly true | 4.4 (0.8) | 273 (90.0) |
| Palliative care is offered only to patients with a cancer diagnosis. | | 1.9 (1.0) | 227 (74.5) |
| Patients are offered palliative care only when there are no other treatments available. | | 3.5 (1.1) | 65 (21.2) |
| Palliative care is also for the patients' family members. | | 3.1 (1.2) | 115 (37.5) |
| Palliative care shortens the patient's life. | | 1.8 (0.9 | 236 (77.6) |
| Palliative care is also offered at the patient's home | | 4.2 (0.7) | 247 (81.1) |
| Palliative care services are only offered in the terminal phase (last days, weeks, or months) of an incurable disease. | | 3.2 (1.2) | 100 (32.9) |
| The costs of palliative care are covered by basic health insurance. | | 3.5 (0.8) | 136 (45) |
| Palliative care is an alternative treatment | | 3.1 (1.2) | 87 (28.3) |
| Palliative care is provided by a team of different specialists. | | 4.1 (0.8) | 245 (80.6) |
| All palliative care is about is pain management. | | 2.7 (1.2) | 156 (51.1) |
| Palliative care also offers social, psychological, and spiritual support to the patient. | | 4.1 (0.8) | 248 (81.5) |
| Palliative care gives a great deal of importance to the patient's preferences (e.g., where to be cured, which therapies to follow). | | 3.8 (0.8) | 205 (67.5) |
| **Attitudes** | | Mean (SD) | |
| I think palliative care services are useful. | 1 = Completely disagree; 7 = Completely agree | 5.9 (1.1) | |
| I would make use of palliative care services, if needed. | | 5.8 (1.4) | |
| I would recommend palliative care services to a family member or a friend, if needed. | | 5.6 (1.4) | |

[1] True beliefs (grey): n (%) "certainly true" or "probably true"; False beliefs (white): n (%) "certainly false or "probably false"

(51.1%). Less than one-third of respondents correctly did not believe that palliative care services are only offered in the terminal phase (last days, weeks, or months) of an incurable disease (32.9%), that it is an alternative treatment (28.3%), and that it is offered only when there are no other treatments available (21.2%).

Overall, men held fewer true beliefs ($p < .05$). No significant gender differences were observed as regards false beliefs, except for the one about palliative care only being offered to cancer patients, to which men believed more than women ($p < .05$). Older respondents correctly believed more often that palliative care is covered by basic health insurance ($p < .001$) and incorrectly believed more often that it is an alternative treatment ($p < .001$). A lower educational level was shown to be significantly associated with a higher tendency to believe several false beliefs ($p < .05$). In contrast, no significant association with education was observed for true ones. No significant associations between beliefs and either nationality or religiosity were observed.

## Attitudes toward palliative care

Respondents showed an overall positive attitude toward palliative care. Details of the answers are shown in Table 2. Considering the sum score, women were shown to hold a significantly

more positive attitude than men ($p < .01$). None of the other socio-demographics were shown to be associated with attitudes.

Linear regression analysis showed that controlling for socio-demographics, personal experience with palliative care, and having a medical background, both self-rated knowledge of palliative care and beliefs were significantly associated with attitudes towards palliative care ($R^2 = .327$, $F(21, 267) = 6.172$, $p < .001$). In particular, a more positive attitude was associated with higher self-rated knowledge (B = .134, $p < .05$) and believing that improving quality of life is among the objectives of palliative care (B = .188, $p < .01$), that palliative care is delivered by an interdisciplinary team (B = .184, $p < .01$), and that palliative care places great importance to the patient's preference (B = .214, $p < .001$). A tendency in the same direction was also observed for believing that palliative care is also offered at the patient's home (B = .107, $p < .10$). On the other hand, believing that palliative care is only for cancer patients tended to be associated with more negative attitudes towards palliative care (B = -.103, $p < .10$).

## Health-related information-seeking, ability, and preferences

Overall, only around one in two respondents reported finding it very or quite easy to find health-related information. The preferred sources for this kind of information were, by far, healthcare professionals (90.7%). Although much less frequently mentioned, other sources relatively often used were websites of cantonal, federal, and international institutions (46.1%), websites of associations or patients' organizations (34.7%), and family and friends (27.1%). Only around one in ten respondents mentioned TV and radio (13.9%), newspapers and magazines (13.5%), or social media and blogs (8.3%) as sources of health-related information.

Slightly more than one out of three respondents was interested in receiving more information about palliative care. When asked about possible channels through which to be informed, the preferred one was a dedicated information point, chosen by more than half of the respondents. Also relatively often mentioned were information through e-mail (47.9%) or regular mail (44.1%), dedicated TV and radio programs (39.6%), and articles in newspapers and magazines (31.0%). Social media, ads in TV and radio, as well as billboards were chosen only by a minority of respondents. More details about information sources and preferences are presented in Table 3. Overall, no significant differences were observed among different socio-demographic groups.

## Discussion and conclusion

### Discussion

This study had the overall objective of informing future communication interventions to promote the utilization of palliative care services in the Italian-speaking region of Switzerland through an in-depth analysis of potential barriers and facilitators. The findings provide us with several important insights into the main factors to address and the best ways to do it.

The first important insight regards the issue of *knowledge*. Lack of public awareness and knowledge about palliative care has long been considered among the main obstacles to using palliative care services, especially in high-resource settings with limited financial and infrastructural barriers to access [22]. Several studies conducted in Switzerland over the last decades have shown that similarly to what happens in other countries, a significant part of the population does not know about palliative care [20, 23, 24], thus sparking several initiatives, such as a National Strategy specifically including dedicated recommendations to address this issue [25]. Our findings, due to these nationwide efforts, show that almost everyone is aware of palliative care. Yet, a more in-depth analysis shows that, in line with studies conducted in other contexts [14–16], this awareness does not translate into a complete understanding of its goals. In

**Table 3. Health-related information-seeking ability and preferences.**

| | | All respondents, N = 313 |
|---|---|---|
| | | n (%) |
| **Ease in finding health-related information** | Very difficult | 1 (0.3) |
| | Quite difficult | 37 (12.0) |
| | Neither difficult nor easy | 104 (33.2) |
| | Quite easy | 143 (45.7) |
| | Very easy | 28 (8.8) |
| **Sources of health-related information** | Healthcare professionals | 284 (90.7) |
| | Websites of cantonal, federal or international institutions | 144 (46.1) |
| | Websites of associations or patients' organizations | 109 (34.7) |
| | Family and friends | 85 (27.1) |
| | TV and radio | 43 (13.9) |
| | Newspapers and magazines | 42 (13.5) |
| | Social media or blogs | 26 (8.3) |
| **Interest in receiving more information about palliative care** | Yes | 110 (35.3) |
| | No | 203 (64.7) |
| **Preferred channels** | Dedicated information point | 58 (52.2) |
| (n = 110) | E-mail (e.g., newsletter) | 53 (47.9) |
| | Flyers in the mail | 49 (44.1) |
| | Dedicated TV and radio programs | 44 (39.6) |
| | Newspapers and magazines | 34 (31.0) |
| | Social media | 14 (12.4) |
| | Ads on TV and radio | 13 (12.2) |
| | Billboards, ads | 6 (5.7) |

particular, our analyses highlight how the knowledge of palliative care is very much limited to pain and symptoms management and the final days of a terminal disease. At the same time, only a few are aware of the potential of palliative care in the early phases of a chronic-degenerative condition and their offer of psychological, social, and spiritual support. This necessarily limits the perceived utility of palliative care services among the public and shows that the association "palliative care = death" is still present in our society. Thus, future interventions must go beyond promoting awareness of palliative care in general and provide a detailed picture of its benefits in the different phases of a disease, detaching the term from the last days of one's life.

A second point, strictly related to the first, regards *beliefs*. An extensive body of literature suggests that false beliefs about palliative care are an essential barrier to the uptake of its services [18]. Our findings indicate that false beliefs are relatively less widespread than true ones. Among the most commonly held false beliefs, we find that palliative care is only for terminally ill patients and is offered only when no other possible treatment exists. On the other hand, we observed that some true beliefs are not very prevalent. For instance, almost one in two respondents did not believe that costs related to palliative care were covered by basic health insurance, and even less believed that palliative care is also directed at family members. If we consider

that, as it will be discussed in more detail later on, holding true beliefs seems to be a more vital determinant of a positive attitude toward palliative care, more efforts are to be put into disseminating correct information on palliative care and showing its benefits rather than into confuting false beliefs. This would also avoid, as the growing body of literature around fake news and conspiracy theories suggests, that false beliefs are involuntarily spread because of the so-called 'familiarity' backfire effect [26].

The third set of considerations regards *attitudes*. Our data show that the public has a positive attitude toward palliative care. This is important, as attitudes are one of the strongest predictors of health-related behaviors [27]. Moreover, attitudes seem more favorable among those who perceive themselves as more knowledgeable and those who hold correct beliefs. Our findings also revealed key beliefs associated with positive attitudes towards palliative care. Specifically, individuals who acknowledge that its goals include improving quality of life, that it is delivered by an interdisciplinary team, that it emphasizes the patient's preferences, and that it can be administered at the patient's home, tended to have a more favorable outlook. It must be noted that negative attitudes were associated with believing that palliative care is only offered to cancer patients, suggesting the need to communicate that this is not the case. This gives us essential insights into the relevant topics to be addressed in communication campaigns and confirms the need to invest in dissemination activities that provide in-depth insights into the aims, target groups, and benefits of palliative care.

The previous points were focused on *what* needs to be addressed. The last two points regard *how* this should be done. First, is *information provision*. Our findings show that around one in three respondents would be interested in receiving more information about palliative care, thus suggesting that an information intervention would be welcome. Also, analyzing health-related information seeking and preferences provides us with some essential insights into how this could occur. We observed that a relatively large group of people found it at least somehow challenging to find health-related information, suggesting they have limited health literacy skills. This stresses the importance of designing informational material that is easily accessible to everyone, for instance, using plain language or complementing written information with illustrations or animations [28–30].

Regarding sources, our respondents showed a strong preference for traditional sources of health information, in particular healthcare professionals. Also, although dedicated coverage in TV/radio or newspapers and magazines was often mentioned and thus should at least be considered an option, the preferred channel was an interpersonal one, namely a dedicated information point. This refers to a centralized resource, either in a physical location or accessible online or via phone, where individuals can obtain comprehensive non-medical information related to palliative care. This resource aims to educate and support patients, families, and caregivers by providing details on palliative services, guidance on navigating care options, legal and ethical considerations, psychological support networks, and practical advice for day-to-day care. This finding might be explained by the topic's sensitive and highly personal nature, better addressed via a one-to-one interpersonal approach rather than anonymously through traditional and new media.

The last point regards *socio-demographics*. Targeting and tailoring messages to the intended audience is one of the pillars of health communication and social marketing [31]. That's why we dedicated specific efforts to identifying possible group differences in the various aspects of interest, which could support delivering a better audience segmentation. However, somehow unexpectedly, no apparent group differences were identified, except for a tendency for people with lower education to hold more false beliefs about palliative care. Besides ensuring to include people with low education and low health literacy levels, in this case, there seems to be no need to develop different communication strategies for different target groups.

We acknowledge that this study is subject to limitations. First, while the LINK Internet Panel provided a broad and diverse sample, we acknowledge a limitation in its accessibility among the elderly population. The requirement for internet use potentially excludes those with limited or no digital access, which is more common among older age groups. This could result in the underrepresentation of elderly viewpoints, particularly significant given their potential need for palliative care. Second, data were collected through a self-administered, anonymous online survey and thus relied on self-reporting rather than objective measures. Moreover, our cross-sectional study design does not provide conclusive evidence about the directions of relationships in the data. Last, like most surveys, there is a risk of self-selection bias [32]. The fact that the people who have agreed to participate in the survey might be most interested in the topic could explain, for instance, the high degree of knowledge on the subject or the overall highly positive attitudes towards palliative care. Overall, however, our findings align with similar studies on the topic in other contexts (e.g., [11, 18]), thus increasing our confidence in their validity.

## Conclusion and practice implications

Palliative care includes health interventions of crucial importance to relieve profound suffering due to degenerative health conditions. On the one hand, palliative care utilization needs to be strengthened. Therefore, community-based initiatives are required to spread awareness and explain palliative care objectives clearly in populations so that people have the correct information to make appropriate choices in partnership with their health professionals. The findings of this study show that health communication interventions are needed and wanted by people because there is still significant unclarity about the goals of palliative care and because it is a field of health where suffering and fears might alter perceptions and attitudes. This study instructs on how to intervene specifically in the Italian-speaking part of Switzerland, including what to communicate and how. But the evidence presented here can inspire similar studies in other linguistic areas of Switzerland and other countries to optimize recognition, knowledge, and understanding of palliative care and to fill gaps in populations' palliative care service demand and utilization.

## Author Contributions

**Conceptualization:** Nicola Diviani, Claudia Gamondi, Piercarlo Saletti, Georg Stüssi, Michel Delbue-Luisoni, Sara Rubinelli.

**Data curation:** Nicola Diviani.

**Formal analysis:** Nicola Diviani, Sara Rubinelli.

**Funding acquisition:** Nicola Diviani, Claudia Gamondi, Piercarlo Saletti, Georg Stüssi, Michel Delbue-Luisoni, Sara Rubinelli.

**Investigation:** Nicola Diviani, Marco Bennardi, Claudia Gamondi, Sara Rubinelli.

**Methodology:** Nicola Diviani, Sara Rubinelli.

**Project administration:** Nicola Diviani, Sara Rubinelli.

**Writing – original draft:** Nicola Diviani.

**Writing – review & editing:** Nicola Diviani, Marco Bennardi, Claudia Gamondi, Piercarlo Saletti, Georg Stüssi, Michel Delbue-Luisoni, Sara Rubinelli.

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
