## [Decision Letter · Decision Letter 0]

7 Nov 2023

PONE-D-22-34909Awareness, Attitudes, and Beliefs about Palliative Care: Results from a Representative Survey of the Italian-Speaking Swiss Population.PLOS ONE

Dear Dr. Diviani,

Thank you for submitting your manuscript to PLOS ONE. After careful consideration, we feel that it has merit but does not fully meet PLOS ONE’s publication criteria as it currently stands. Therefore, we invite you to submit a revised version of the manuscript that addresses the points raised during the review process.

We look forward to receiving your revised manuscript.

Kind regards,

Laura Brunelli, MD, PhD

Academic Editor

PLOS ONE

Journal Requirements:

a) Did participants provide their written or verbal informed consent to participate in this study?

Reviewers' comments:

Reviewer's Responses to Questions

**Comments to the Author**

1. Is the manuscript technically sound, and do the data support the conclusions?

Reviewer #1: Yes

Reviewer #2: Yes

2. Has the statistical analysis been performed appropriately and rigorously? 

Reviewer #1: Yes

Reviewer #2: Yes

3. Have the authors made all data underlying the findings in their manuscript fully available?

Reviewer #1: Yes

Reviewer #2: Yes

4. Is the manuscript presented in an intelligible fashion and written in standard English?

Reviewer #1: Yes

Reviewer #2: Yes

5. Review Comments to the Author

Reviewer #1: Thank you for giving me the opportunity to review this well-written and relevant article - the results are very interesting and clearly described.

I have a few comments:

- the authors use "PC" aas abbreviation without prior explanation - please add or revise

- some sentences are rather long or complicated (e.g. chapter 1 Introduction 3rd paragraph starting with "in addition...", chapter 4.1 4th paragraph starting with "the third set of considerations... 2nd sentence is rather long, 5th paragraph starting with "the previous points..." 2nd sentence why "instead"? - please revise

- ad 2.1 sample: please give some more information on how this "internet panel" works and please dicscuss the limitation of such a population because especially the elderly population might not have access to this panel which might limit the interpretation

ad 2.2.1. - if respondents never heard of the term - where they redirected to the socio-demographic section or the health-related information-seeking section? why where they not asked to fill in section iv? also - could the authors state why they chose the "facts and (false) beliefs which they mentioned - there might be others too and the authors made a selection

ad 2.2.5. - why did you ask about "religiosity" - this was not mentioned as potentially relevant in your introduction? Religiosity was also not analysed as relevant factor (compared to gender, ...) - was it in any way relevant?

ad 2.3 - based on which methodological approach did the authors perform the thematic analysis?

- ad 4.1 discussion: could you explain more, what a "dedicated information point" is? could the authors add the references to the last sentence in the discussion in which they refer to "...similar studies on the topic..."

- Table 3: I would find it more helpful if the authors could sort the answers to questions 2-4 based on the frequency. Just out of curiosity: what are the thoughts of the authors that most respondents did not wish any additional information about palliative care?

- could the authors add specific details to the references e.g. as to reference 1 (access specifics are needed)

Reviewer #2: I don't have other comments

6. PLOS authors have the option to publish the peer review history of their article (what does this mean?). If published, this will include your full peer review and any attached files.

Reviewer #1: No

Reviewer #2: No

---

## [Author Response · Author response to Decision Letter 0]

9 Nov 2023

A detailed response to the reviewers' comments has been uploaded into the system.

---

## [Editor Report · Decision Letter 1]

10 Nov 2023

Awareness, Attitudes, and Beliefs about Palliative Care: Results from a Representative Survey of the Italian-Speaking Swiss Population.

PONE-D-22-34909R1

Dear Dr. Diviani,

We’re pleased to inform you that your manuscript has been judged scientifically suitable for publication and will be formally accepted for publication once it meets all outstanding technical requirements.

Kind regards,

Laura Brunelli, MD, PhD

Academic Editor

PLOS ONE

---

## [Editor Report · Acceptance letter]

16 Nov 2023

PONE-D-22-34909R1 

Awareness, attitudes, and beliefs about palliative care: Results from a representative survey of the Italian-speaking Swiss population. 

Dear Dr. Diviani:

I'm pleased to inform you that your manuscript has been deemed suitable for publication in PLOS ONE. Congratulations! Your manuscript is now with our production department. 

Kind regards, 

on behalf of

Dr. Laura Brunelli 

Academic Editor

PLOS ONE